# Factors Affecting the Nutritional, Health, and Technological Quality of Durum Wheat for Pasta-Making: A Systematic Literature Review

**DOI:** 10.3390/plants12030530

**Published:** 2023-01-24

**Authors:** Silvia Zingale, Alfio Spina, Carlo Ingrao, Biagio Fallico, Giuseppe Timpanaro, Umberto Anastasi, Paolo Guarnaccia

**Affiliations:** 1Department of Agriculture, Food and Environment (Di3A), University of Catania, Via S. Sofia n. 100, 95123 Catania, Italy; 2Agricultural Research Council and Economics (CREA)—Research Centre for Cereal and Industrial Crops, Corso Savoia, 190, 95024 Acireale, Italy; 3Department of Economics, Management and Business Law, University of Bari Aldo Moro, Largo Abbazia Santa Scolastica, 53, 70124 Bari, Italy

**Keywords:** semolina, pasta, phytochemicals, nutrients, agronomic management, processing, genotype

## Abstract

Durum wheat is one of the most important food sources in the world, playing a key role in human nutrition, as well as in the economy of the different countries in which its production areas are concentrated. Its grain also represents a staple and highly versatile ingredient in the development of health foods. Nonetheless, the aspects determining durum wheat’s health quality and their interactions are many, complex, and not entirely known. Therefore, the present systematic literature review aims at advancing the understanding of the relationships among nutritional, health, and technological properties of durum wheat grain, semolina, and pasta, by evaluating the factors that, either positively or negatively, can affect the quality of the products. Scopus, Science Direct, and Web of Science databases were systematically searched utilising sets of keywords following the PRISMA guidelines, and the relevant results of the definitive 154 eligible studies were presented and discussed. Thus, the review identified the most promising strategies to improve durum wheat quality and highlighted the importance of adopting multidisciplinary approaches for such purposes.

## 1. Introduction

Durum wheat (*Triticum turgidum* L. subsp. *durum* (Desf.) Husn) is a tetraploid species (2n = 28) farmed annually on an estimated cropping area of 18 × 10^6^ ha, indicative of nearly 8–10% of the total wheat cultivated area in the world, and a yearly production varying from 35 to 40 × 10^6^ Mg [1]. Generally, it is well suited to locations with low and/or irregular rainfall and repeated heat stress, such as the Mediterranean Basin, where it represents a staple crop and a valuable commodity [2]. Other major durum wheat producers, besides the countries of the Mediterranean Basin (Turkey, Italy, Morocco, Syria, Tunisia, France, Spain, and Greece), are Canada, Mexico, USA, Russia, Kazakhstan, Azerbaijan, and India, with the first three being the most important exporters [2]. With a cropping area of 1.3 × 10^6^ ha, Italy is the top producer of durum wheat in the EU, providing 4.1 × 10^6^ Mg of grain [3]. 

Compared with common wheat (*Triticum aestivum* L. subsp. *aestivum*), which is the largest cultivated cereal crop, durum wheat is distinguished by stronger gluten, grain yellow colour, and longer durability, all of which are necessary for pasta-making [4]. The latter, however, does not correspond to the exclusive end-product because bread made with durum wheat semolina and pasta made with common wheat flour also exist [5]. Regarding this, Italy has a considerable role in durum wheat production, mostly as a result of the economic status of the pasta industry, which has pushed the intense breeding work conducted since the beginning of the 20th century [6].

Thus far, it has not been possible to propose a simple and exhaustive definition of durum wheat quality because it changes depending on the supply chain’s stakeholders (farmer, grain dealer, seed company, milling industry, pasta industry, consumer), and end-use [7]. Nevertheless, the quality of final products is influenced by the quality of the grain and semolina, which, in turn, are mainly controlled by the genotype but also by the environmental conditions, crop management, and processing techniques. In this regard, Sicignano et al. 2015 [8] presented a step-by-step manual to simplify the understanding of the most important issues that can affect pasta characteristics. 

It is particularly well recognised that the main contributing factors to pasta quality are protein content and gluten properties [7], which are responsible for dough properties such as dough strength, extensibility, and stability. Indeed, cooked pasta has a distinct structure, consisting of a protein network that firmly entraps starch granules. The structure and composition (amylose content and ratio of large to small starch granules) of such a matrix contribute to the nutritional properties of pasta and make the latter contain slowly digestible starch and have a medium-low glycaemic index [9,10,11]. Accordingly, consuming pasta was associated with lower postprandial blood glucose and insulin response and with decreased esophageal cancer risk when compared with bread [12,13].

High levels of yellow pigments and lesser oxidative enzyme activity are additional significant features to consider when making pasta with superior cooking and sensory properties [14].

Pasta may also be a means of phytochemicals, such as phenolic compounds, vitamins, and minerals, which contribute to health benefits, depending on the raw material used in its preparation. However, reduced information on the content and composition of such bioactive compounds in durum wheat is available [15]. Indeed, current cereal breeding programs have addressed mainly higher yields and improved technological quality for industrial processing whilst neglecting traits related to nutrition, health, digestibility, and potential allergenicity [16,17]. From this point of view, recent studies on the phytochemicals content of durum wheat products have recently been conducted, underscoring very attractive opportunities for breeding programs to improve wheat nutritional and functional quality [18,19,20,21,22]. Other trends regarded the incorporation of plant-based ingredients, probiotic strains, and/or fibre into pasta products to improve their nutritional value and associated health benefits. However, more research still has to be conducted to evaluate other beneficial compounds in cooked pasta and to estimate more deeply the conditions of the pasta-making process that strongly impact the quality of final products [9].

In virtue of these considerations, the aim of this systematic literature review was to advance the understanding of nutritional, technological, and health properties of durum wheat grain, semolina, and pasta, through an integrated and holistic approach. To that end, this paper’s authors evaluated all those factors that could, either positively or negatively, affect the quality of the durum-wheat-derived products, such as semolina and pasta. In this respect, according to the author’s knowledge, this is the first time that a systematic literature review (SLR) has been performed on so many quality aspects of durum wheat. In fact, other reviews provided valuable insights on a single or small number of durum wheat quality-related topics, such as breeding history and quality [6,23,24,25,26], human health effects of durum wheat consumption [27,28], enrichment of pasta with non-conventional ingredients [29,30], grain composition and end-use quality [31,32], pasta-making [8,33,34] and drying impacts [35,36], and durum wheat colour [37,38], etc. This might be interpreted as a sign of the novelty of this paper and the added value it can bring to the literature by identifying new research priorities and innovative approaches to enhance the overall quality of durum wheat for pasta production.

## 2. Materials and Methods

This systematic review was conducted following the PRISMA guidelines [39]. To perform the literature search, Science Direct, Scopus, and Web of Science were screened to retrieve published studies regarding the factors affecting the nutritional, technological, and health properties of durum wheat grain, semolina, and pasta. The following search keywords were used: (“durum wheat”) AND (grain OR semolina pasta) AND (nutritional OR technological OR health quality) AND (effect OR influence OR impact). The above-mentioned databases were chosen for their research builder’s simplicity and wide selection of Life Sciences Journals featuring original research, insightful analysis, current theory, and more.

The obtained articles were screened in a two-step selection process to ensure their relevance:In the first step, authors read titles and abstracts and excluded those articles that did not explicitly examine the effects of one or more specific variables/factors on the technological, nutritional, and/or health properties of durum wheat samples of grain, semolina and pasta;In the second step, the full texts of the selected articles were critically read, and then were discarded those that (a) did not include original research data; (b) had not implemented appropriate treatments, or had not used accurate analytic methods to simulate and measure, respectively, the effects of the investigated factor on the quality properties of durum wheat; (c) had not been peer-reviewed; and (d) had not been entirely published in English. Thus, the authors did not include review and meta-analysis articles, book chapters, opinion/commentary, and conference papers.

Exhaustive details of the screening process can be found in the PRISMA flow chart (Figure 1). The selected articles were classified concerning details of publication (authors, title, year of publication) and the type of variables/factors investigated.

## 3. Results and Discussion

### 3.1. Overview of the Reviewed Studies

Among the studies published until 26 June 2022 (data last searched), 257 articles were found from the Scopus (86), Science Direct (31), and Web of Science (140) databases. After removing duplicates, 220 research articles were identified. Then, according to titles and abstracts, 171 papers were recognised as potentially suitable, whereas 49 papers were excluded due to the presence of irrelevant content or information on other qualitative aspects of durum wheat in titles and abstracts. Therefore, the full texts of the remaining 171 papers were scrutinised, and finally, 154 eligible articles were included in the qualitative synthesis. Lists of references (cross-referencing) were also examined to identify relevant studies to complete the search, and 21 additional articles were included.

Figure 2 shows the annual scientific production dealing with the research on factors affecting durum wheat quality: from it, it is possible to note that the number of papers has fluctuated over the last two decades, reaching a maximum of 17 papers in 2019, 2020, and 2021.

Readers are referred to Figure 3 and Table 1 for a summary of the variables investigated within the studies and to Section 3.2, Section 3.3 and Section 3.4 for a more critical and exhaustive discussion of the objectives, findings, and recommendations of the different reviewed studies.

As shown in both Figure 3 and Table 1, research on DW quality has been mainly focused on the following: -The experimentation with new pasta formulations;-The impacts of the processing;-The influence of agronomic management;-The selection of genotypes.

In contrast, the effects of environmental stresses and G × E × AM interactions have been poorly studied.

Moreover, as indicated by the study periods in Table 1, the majority of the factors have been investigated within the last two decades, except for processing ones with a longer history of research, as documented by the first paper published in 1989.

This could be explained considering the interest of both researchers and industry in identifying, through scientifically sound methods, the key elements to improve semolina and pasta quality traits.

### 3.2. Cropping Factors

#### 3.2.1. Genotype Effects

This section was aimed at reviewing papers that evaluated the genotype effects on durum wheat’s technological, nutritional, and health-promoting properties. Indeed, wheat breeding programs were mainly aimed at the development of wheat varieties that provide superior-quality finished products, especially in relation to technological aspects. However, because customers are paying more attention to their health, wheat breeding projects are increasingly designed for enhancing production and endowing derived products with a higher nutritional value [27]. As a consequence, a number of studies have investigated how G, E, and G × E interactions can influence the expression of different quality features, such as: Nutritional and technological properties;Other quality features determined by the accumulation of certain metabolites [103].

Specifically, a significant process of genetic erosion has been triggered by the replacement of wild relatives and landraces with improved varieties, resulting in the loss of possibly beneficial unexplored alleles. The latter could be especially advantageous to improve the currently cultivated durum wheat genotypes’ ability to tolerate biotic and abiotic stresses, as well as their nutritional and technological quality [185]. Consequently, scientists are devoting increasing resources and work to the search for beneficial alleles and traits from local germplasm collections in order to restore their programs toward food security and quality [186].

Such a trend was upheld by this review’s findings since many of the considered studies were found to be focused on the screening and/or comparison of durum wheat germplasms, including landraces, old, and modern varieties, related to different quality aspects. For instance, several studies evaluated the genotype effects on **durum wheat protein quantity and quality** [44,45,47,48]. Indeed, the latter has always been among the crucial subjects of wheat breeding, with applications mostly focused on the selection of genotypes associated with strong gluten, generally measured with gluten index or rheological tests [48,187]. In their recent two-year field study, De Santis et al. 2017 [47] carried out a comparative assessment of eight current and seven old Italian durum wheat genotypes to examine the changes in gluten composition and quality raised by the breeding of Italian durum wheat genotypes in the 20th century. Through the discussion of the results, the authors stated that modern genotypes were characterised by a higher gluten quality compared with old cultivars, as measured through the gluten index, and attributed that fact to a significantly major expression of B-type low molecular weight glutenin sub-units (LMW-GS), and a higher glutenin/gliadin ratio. Moreover, there were no significant differences between old and modern genotypes relating to α-type and γ-type gliadins, the former of which is regarded as one of the major fractions determining coeliac disease.

In line with this, De Santis et al. 2018 [48] also evaluated the grain protein composition of two groups of durum wheat varieties, namely old and modern ones, under Mediterranean environment conditions. As for De Santis et al. 2017 [47], in this case, modern genotypes revealed a higher relative content of soluble glutenin and a slight decrease in the amount of gliadin fraction, which is generally responsible for gluten-related disorders.

Similarly, Ficco et al. 2019 [44] assessed gluten peptides release during in vitro digestion as well as the content of potential prebiotic carbohydrates in nine old and three modern durum wheat genotypes. According to their findings, the old genotypes’ whole meal flours are unable to cause a lower exposure to gluten peptides and, in many cases, produce a greater amount of immunogenic and toxic peptides than modern varieties. Regarding possibly prebiotic carbohydrates, instead, even though the modern varieties were richer in total starch, the old genotypes featured a greater amount of resistant starch, always > 1%.

Rises in gluten strength of modern varieties through time were also evaluated by Roncallo 2021 [46] in a worldwide durum wheat collection and were closely associated with changes in allele composition in the high molecular weight glutenin sub-units (HMW-GS) and LMW-GS, specifically with a progressive replacement of the 20x + 20y banding pattern by the 7 + 8 and 6 + 8 subunits at Glu-B1.

Likewise, another work within those reviewed, namely Fiore 2019 [55], was aimed at studying the germplasm of durum wheat Sicilian landraces by measuring its agronomic, morphological, and quality features and utilising single nucleotide polymorphisms (SNP) markers. As expectable, within the quality-related traits, they discovered significantly greater protein content (always >16.0%) and weaker gluten (as evidenced by gluten index values ranging from 18.8 to 59.5) in the Sicilian landraces compared with two modern cultivars. Moreover, the protein content and the characteristics related to gluten (dry and wet gluten, water binding in wet gluten, and gluten index) were also the quality traits with the stronger influence in discriminating landraces from the other groups of cultivars used as testers, namely a bread wheat variety, two traditional varieties, and two modern ones.

From that perspective, the dilution effect of proteins and other minor components for the increased amount of carbohydrates may be another possible cause for the decrease in protein content in durum wheat varieties, in addition to the direct genetic effects of the breeding conducted up-to-date [49]. Hence, the pasta-making industry has to blend different cultivars with high gluten quality to reach proper processing properties. Padalino et al. 2014 [49] emphasised that the decrease in protein concentration is a real issue for this sector. According to them, even though it is possible to state that these cultivar mixtures could represent a good compromise for the technological quality aspects, the same is not completely true for the nutritional attributes.

To investigate this position, Padalino et al. 2014 [49] studied six distinct mono-varietal durum wheat cultivars (both modern and old varieties) and evaluated their effects on pasta quality compared with a control pasta made with a commercially available semolina mixture. According to their results, the Cappelli (old variety) pasta performed the highest protein and ash contents; the lowest available carbohydrate content; the lowest glycaemic index; and higher firmness, adhesiveness, and bulkiness values compared with the other spaghetti samples. Therefore, the authors concluded that semolina products based on mono-varietal cultivars could provide durum wheat pasta with a good balance between nutritional and cooking quality.

How genetics can influence gluten quality by determining differences in the amount and composition of glutenins, gliadins, and **celiac disease (CD)-triggering peptides** was also investigated. In particular, one study among those reviewed, Ronga et al. 2020 [113], analysed the accumulation of gluten-derived toxic and immunogenic peptides using proteomic techniques in six durum wheat genotypes belonging to the Italian Durum Wheat Network of variety trials. The results showed that whilst toxic peptides’ accumulation was heavily influenced by genotypes, immunogenic peptides expressed marked variation across the different locations, with significant genotype-by-year and genotype-by-site interactions. Moreover, a weak association between grain yield and protein content and toxic and immunogenic peptide accumulation was detected. Overall, the conclusions by Ronga et al. 2020 [113] were central to specifying the possibility of selecting the agronomic conditions to maximise production in terms of yield, maintain good protein content whilst decreasing the content of peptides implicated in the CD, and consequently reduce the exposure of non-celiac predisposed subjects.

Two of the studies that were reviewed—namely, those by Hogg et al. 2015 [50] and Botticella et al. 2016 [51]—introduced another way to obtain healthier pasta with a low glycemic impact. Both papers assessed the effects of SSIIa null mutations on durum wheat grain and pasta quality, as these kinds of mutations are expected to significantly increase **the amylose content**, thus lowering the glycemic index of pasta and enhancing its quality characteristics. Indeed, the extent of digestibility of starches generally decreases as amylose content increases, although amylose content alone is not the sole predictor of digestibility [188]. According to Botticella et al. 2016 [51], the SSIIa null semolina exhibits novel rheological behavior and an elevated content of all major dietary fibre components, including arabinoxylans, β-glucans, and resistant starch. Similarly, Hogg et al. 2015 [50] assessed the quality technological and sensory characteristics of pasta and discovered that high-amylose line results in lower water absorption, increased cooking loss, shorter cook time, and considerably higher firmness, even after overcooking.

In addition to proteins and starch, the bright yellow **colour** of durum wheat products is another crucial quality attribute since it appeals to the consumer and has nutritional significance [42]. Clarke et al. 2006 [40], one of the studies reviewed here, provided interesting and useful insights about the inheritance of grain or semolina pigment concentration: specifically, their study evaluated seven durum wheat crosses of high by low pigment concentration parents under five field experiments carried out at two or more locations in Western Canada for 2 or more years. Following their findings, the pigment concentration varies depending on the environment, and the inheritance is multigenic and strong when tested in replicated, multi-location, multiyear experiments. This gave Clarke et al. 2006 [40] the opportunity to suggest employing additional genetic tools, such as DNA markers, to improve the selection of parents for breeding programs.

Another new breeding target, such as the esterification ability for greater retention and accumulation of carotenoids in durum wheat, was suggested by Requena-Ramírez et al. 2021 [43]. The latter identified different accessions of the Spanish durum wheat landraces germplasm that could offer good chances for carotenoid biofortification projects.

Moreover, as underlined by Ficco et al. 2019 [44], a new interest in breeding has also been shown for another class of pigments, such as anthocyanins, because of their antioxidant potential. In contrast to yellow pigment, Ficco et al. 2019 [44] found that anthocyanins expression is highly hereditary, with genotype × year interactions having a less significant impact. Future programs might benefit from this characteristic to improve the antioxidant properties of wheat-derived food products.

As previously reported above, genotype effects on the contents of **micronutrients and bioactive compounds were also studied**. That is why, although durum wheat primarily serves as a source of carbohydrates and, to a less extent, of proteins, its consumption in the form of wholemeal durum-based products can potentially contribute to enriching human dietary habits with a wide range of beneficial compounds [52]. Micronutrients, dietary fibre, and phenolic acids are only a few of the **bioactive components** that are mostly contained in the grain bran and germ tissues [15].

**Mineral elements**, for instance, have been proposed to have a vital role in maintaining good health, but they are complexed with phytic acid on kernels and, as a result, are not nutritionally available. Therefore, the selection of cultivars with high mineral density and low phytate content in mature seeds should be a focus of breeding strategies to increase the nutritional value of durum wheat products [52].

In order to address this, Velu et al. 2017 [52] examined the variation in grain mineral cations (Ca, K, Mg, Mn, Na, Cu, Zn, and Fe) and P concentrations in a group of durum wheat genotypes representative of the Mediterranean old and modern germplasms. Specifically, they assessed how dwarfing genes affected grain Zn, Fe, Mn, and Mg contents, as well as kernel weight. The results allowed Velu et al. 2017 [52] to observe that the dwarfing genes were associated with a decrease in all the minerals and with a negative effect on kernel weight. In particular, Velu et al. 2017 [52] concluded that such changes in micronutrient concentrations are mostly caused by the associated pleiotropic effects of dwarfing genes on increased biomass partitioning and higher harvest index. Such dilution of minerals concentration as yield potential rises highlighted the difficulty of using wild relatives to improve modern varieties and, therefore, the need to screen landraces, which tend to have a better agronomic type than wild relatives [189,190].

Within the reviewed literature, the research performed by Hernandez-Espinosa et al. 2020 [15] was precisely in this direction. These authors, indeed, examined the contents of arabinoxylan, phenolic acids, phytic acid, and micronutrients in a sample of eighty-two durum wheat landraces (thirty-nine Iranian and forty-three Mexican). Their findings showed that landraces varied widely in all of the quality traits, with some cultivars being characterised by a desirable combination of such attributes. The latter landraces were proposed as a useful genetic resource to improve nutritional quality, particularly the quantity and bioavailability of dietary fibres and micronutrients.

Regarding **dietary fibres**, important findings concerning the role of genotype on the arabinoxylans content were reported by Ciccoritti et al. 2011 [106]. Their experiment was carried out with a large collection of Italian durum wheat cultivars grown in two different Italian sites for two years to assess the influence of genotype and environmental factors on the arabinoxylans’ content and extractability. Their findings confirmed the genetic control of arabinoxylans’ structural variability and differences among genotypes for arabinoxylan fractions. Moreover, the environment and its interaction with genotype appeared to be important sources of variation for arabinoxylans in the whole grain. However, the combined effect (G × E) contributed to the total variability in a lower way compared with genotype and environment alone.

Furthermore, a number of studies considered in this review were focalised on the genotype effects and/or genotype x environment effects on **phenolic compounds**’ content and quality [18,19,20,58,60,104,107,108,111,112]. Specifically, the interest in such grain phytochemicals is linked to their strong antiradical activity, which can contribute to protecting humans against various degenerative diseases [20].

For instance, the variability for individual and total phenolic acids contents within a large set of tetraploid wheat, including the *Triticum* subspecies *durum, turanicum, polonicum, carthlicum, dicoccum*, and *dicoccoides*, was recently assessed by Laddomada et al. 2017 [58]. Compared with other subspecies, durum wheat cultivars have higher average contents of total phenolic acids (from 550.9 to 1701.2 lg g^−1^ dm), which were also more stable over time. Interestingly, they found a significant genotype effect for both total and individual phenolic acids, as well as a moderately high ratio of genotypic variance to the total variance, indicating that a breeding approach to increase phenolic acids concentration in durum wheat could be taken into consideration.

Moreover, Martini et al. 2015 [112] assessed the level of total antioxidant capacity in ten durum wheat genotypes grown in the same experimental field over three successive cropping years. They also measured the content of phenolic acids and of total phenolics, both occurring as soluble free, soluble conjugated, and insoluble bound forms, as well as the yellow-coloured pigments. The findings of their study highlighted that in contrast to the content of phenolic acids and total phenolics, which appeared to be most affected by the environment and then by the genotype, yellow-coloured pigments and total antioxidant activity are mainly influenced by genetic factors. This called for programs aiming at selecting genotypes naturally rich in antioxidant compounds, also by choice of the more suitable growing area.

In line with this, Dinelli et al. 2009 [20], Dinelli et al. 2013 [19], Di Loreto et al. 2018 [18], and Truzzi et al. 2020 [60] determined the qualitative profiles of phenolic compounds (free and bound fractions) in diverse wheat, comprised of old and modern durum wheat genotypes.

Dinelli et al. 2009 [20] evaluated such grains’ functional components in two old and seven modern Italian cultivars of durum wheat cropped in the same location and years. No significant differences were detected among the investigated cultivars as regards the amounts of total phenolic and flavonoid compounds, but a remarkably qualitative diversity in the phytochemical profiles between old and modern varieties was noted: in particular, the mean numbers of total compounds and total isomers were significantly higher in old genotypes than in modern ones. Thus, landraces were indicated as a rich source of genetic diversity, especially for bioactive compounds, allowing the authors to suggest their uses in the formulation of both standard and specialty functional food products.

Along with the phytochemical composition, Dinelli et al. 2013 [19] also assessed the agronomic performances and the nutritional value of several old and modern durum wheat genotypes grown under low input farming conditions. No clear distinction between the two groups of varieties was observed in terms of grain yield, demonstrating how the productivity gap, often existing between dwarf and non-dwarf genotypes, is significantly narrowed when they are managed under low inputs. All the examined genotypes also had high protein amounts and provided phytochemicals such as dietary fibre, polyphenols, flavonoids, and carotenoids. On the other hand, Di Loreto et al. 2018 [18] discovered remarkable quantitative variations between old and modern genotypes for their phenolic acids content and composition, as well as for the antioxidant activities, with significantly higher values obtained for the old ones. Furthermore, since the varieties were grown in the same area and year, the observed differences were to be attributed to a genetic influence. In light of this, the authors hypothesised that old and modern durum wheat genotypes might have different secondary metabolite biosynthesis pathways. When it comes to the bound forms of polyphenols and flavonoids, as well as to the antioxidant activity, such genetic control was also supported by the results reported in Di Silvestro 2017 [107], although a significant interaction between cropping year and location was found for the free forms of phenolics. In line with this, Bellato et al. 2013 [104] found that the environment and genotype were the primary determinants of the phytochemical profiles of durum wheat grains, with their interactions (E × G) playing a much lower role in the overall variability.

Following Truzzi et al. 2020 [60], this research line, which aims at distinguishing wheat varieties based on their health-promoting compounds profiles, may open the way and support new approaches for the genetic improvement of the genus *Triticum* and the development of nutraceutical cereal-based ingredients and products.

In this regard, Truzzi et al. 2020 [60] combined the assessment of the polyphenol content and antioxidant activity with the use of cell models systems, where the polyphenols extracted from different soft and durum wheat genotypes involving modern and old varieties were tested for their ability to repair intestinal tissue damages. The results showed differences between soft and durum wheat phenolic compounds, implying a significant genotype influence on the proprieties of wheat extracts. Moreover, both modern and old varieties’ polyphenols had positive effects on cells. That is why Truzzi et al. 2020 [60] concluded that combining old and modern varieties in specific crosses could be a useful strategy for developing nutritionally improved wheat-based products to be integrated into diets with protective functions against chronic and inflammatory diseases. Nevertheless, human intervention studies are required to validate these results and possibly support health claims regarding the functional properties of wheat varieties.

Finally, the genotype can also influence the **volatile composition** of durum wheat kernels with consequences on semolina and pasta products [53,54]. Both Beleggia et al. 2009 [54] and Mattiolo et al. 2017 [53] found considerable differences in terms of composition and amount of volatile compounds (especially of ketones, aldehydes, and alcohols) among durum wheat cultivars, using headspace solid-phase micro-extraction (HS-SPME) and the GC–MS. Such research opened up the selection of varieties based on the typical flavour they could give to the respective monovarietal pasta and pointed out the efficacy of HS-SPME in conjunction with GC-MS for such purposes.

#### 3.2.2. Environmental Effects

In this section, the influence of different **environmental stresses** is widely and critically described through the evaluation of the reviewed studies’ results. Additionally, the exacerbation of environmental stresses due to climatic change underway is expected to heavily affect not only crops’ productivity but also the chemical composition and, thus, the quality of the products, potentially leading to threats to food security [66,191]. This is even more evident for durum wheat, as its areas of production are characterised by drought and extreme temperatures [192,193,194].

Durum wheat’s quality features are generally strongly influenced by **environmental conditions**, with growing zones, latitudes, and moisture regimes showing the greatest impacts [44]. For instance, one study suggested the possibility that the expected values of gluten index and colour parameter b* (CIELAB System) related to yellow, particularly important for pasta-making, could be accurately estimated based on meteorological variables, such as rainfall, mean temperature, and the number of heat days [43].

Among environmental stresses, the impact of **high temperature during grain filling** on the qualitative characteristics of wheat has been long recognised [46].

The first paper dealing with this topic among those reviewed was that of Corbellini et al. 1997 [46], who found that the occurrence of very high temperatures (in the range of 35–40 °C) during grain filling substantially affects dry matter and protein accumulation in the different parts of the plant. Specifically, according to their results, heat stresses that came late in grain filling determined a “dough weakening” effect, which may reduce the commercial value of the production.

However, there exist different typologies of heat stresses during wheat grain filling, and they can give rise to different effects on durum wheat quality, depending on the timing (days after anthesis) and duration.

Recently Sissons et al. 2018 [50] pointed out that late sowing (two months later than normal sowing) had positive effects on increasing protein content, dough quality indicators, and pasta end-use traits since it exposed the crop to more days at higher temperatures (around 30 °C). However, the same practice determined a reduced yield, grain weight, test weight, and milling yield [50].

In line with this, Cosentino et al. 2019 [70] investigated the effect of post-anthesis heat stress on several durum wheat genotypes, including old and modern varieties and a Sicilian landrace. According to their results, advances in main phenophases significantly decreased kernel weight and grain yield, whereas grain protein content increased. Moreover, genotypes responded differently to heat stress, as evidenced by the net photosynthesis, transpiration rate, instantaneous water use efficiency, and dry matter accumulation in kernels.

In addition to the influence on protein quantity and quality, heat stress can also have strong effects on grain composition in terms of the compounds that are beneficial or detrimental to human health. In this regard, an interesting study was carried out by de Leonardis et al. 2015 [71], looking at how the nutritional value, antioxidant capacity, and metabolic profile of different durum wheat genotypes can be affected by heat stress, in particular by the exposure at 37 °C for five days after the flowering stage. Following them, heat-stress effects can have diverse consequences on the accumulation of bioactive compounds, which can result in either beneficial or detrimental effects on human health in relation to the wheat genotype and the environmental conditions that occur during the crop cycle.

In line with Sissons et al. [50] and de Leonardis et al. 2015 [71], Rascio et al. 2015 [69] also evaluated the effects of heat stress on the potentially health-beneficial compounds of two old durum wheat genotypes and a modern cultivar, sown in winter and spring. As for de Leonardis et al. 2015 [71], the results evidenced that genotypic effects had the most significant role in the determination of grain quality and that the genotypes do not always react the same way to environmental changes. Specifically, spring sowing changed the grain composition of polyphenols and further differentiated the genotypes in terms of carotenoids, high-quality starch, and sulfur compounds, with variations depending on the genotype. Consequently, the timing of sowing could be changed to achieve the desired enrichment of cereal-based foods according to the end-use and type of health benefits required by consumers [69].

There is less research on the impacts of drought durum wheat quality than there is on heat stress, whether it is caused by erratic or deficient rainfall or limited irrigation [67]. Among the reviewed articles, Flagella et al. 2010 [72] and Giuliani et al. 2015 [55] focused on water deficit or drought stress effects on durum wheat protein composition. Investigating that was particularly important because the strength and elasticity of the wheat are strongly influenced by the insoluble fraction of wheat grain protein, which is made up of gliadins and glutenins. Specifically, Flagella et al. 2010 [72] found that the timing of stress occurrence varied the effects of water deficit on the technological quality and protein composition. When a water deficit persisted through the growing season, an increase in protein content and a drop in the HMW-GS/LMW-GS ratio were detected by the authors. Instead, when terminal water stress occurred in grain filling, an improvement in gluten strength was observed. Consistent findings were also determined in the study carried out by Giuliani et al. 2015 [55], who used a proteomic approach to evaluate changes in storage protein composition under water stress of two Italian modern durum wheat cultivars: in particular, LMW glutenin subunits decreased under water stress, whilst gliadin-like (especially α-gliadins) proteins increased.

Furthermore, some studies among those reviewed focused on the **combined effects of heat and drought stress** on several nutritional and technological characteristics of durum wheat [65,67,109,195].

Specifically, Li et al. 2013 [67] examined the effects of heat and drought stresses on the technological durum wheat quality of nine different cultivars, each with a different set of quality attributes. According to their findings, drought tends to enhance gluten strength, whereas heat stress tends to reduce it. Moreover, as in Corbellini et al. 1997 [46], a “weakening” effect of heat stress on the gluten strength of durum wheat was reported. Moreover, Gagliardi et al. 2020 [109] evaluated the variation in the gluten protein assembly of four durum wheat genotypes with respect to various nitrogen levels and growing seasons. Significant lower yield and a higher protein concentration were observed in the year characterised by a higher temperature at the end of the crop cycle. Moreover, the effect of the high temperatures on protein assembly was different for the genotypes in relation to their earliness. Phakela et al. 2021 [65], instead, aimed at assessing the effects of drought and heat stress (both in moderate and severe ways) on the variation in gluten proteins in six durum wheat cultivars with the same HMW and LMW glutenin subunit composition. Overall, Phakela et al. 2021 [65] observed that sulfur-rich proteins (α/ β- and γ-gliadins, and LMW glutenins) were more sensitive to heat and drought conditions, except for the α-gliadins, which were significantly increased by all stress treatments. Moreover, in their study, the γ-gliadin, which was highly influenced by genotype, has a positive effect on alveograph characteristics under all conditions, thus indicating that selection for γ-gliadin will be successful in increasing dough strength.

Recently, Erice et al. 2019 [49] simulated future climatic conditions, consisting of **elevated CO_2_ and drought**, to test their effects on grain quality features and water usage efficiency of two varieties of durum wheat, namely a landrace and a modern variety. The authors found that the two cultivars showed opposite behavior regarding kernel quality and water use efficiency when exposed to the projected future atmospheric CO_2_ levels. Both varieties provided similar protein amounts when cultivated under ambient CO_2_ and full water availability, whilst they responded differently to elevated CO_2_. In particular, the modern genotype’s protein content decreased in both water regimes, whilst the landrace’s one did not change. On the other hand, better agronomic performances and technological properties were observed for the modern variety under the projected future elevated CO_2_ and drought conditions. Despite this, landraces were proposed to be incorporated into breeding programs due to their lower susceptibility to losing the kernel protein nutritional value [66].

Equally, Soba et al. 2019 [56] revealed that important modifications of grain metabolism due to CO_2_ emissions increases could have implications for its nutritional quality. Indeed, the growth of durum wheat under elevated CO_2_ conditions reduced the N content as well as the protein and free amino acids content, with a remarkable loss in glutamine, which is the most prevalent amino acid in grain. Indeed, approximately 40% of gluten proteins is represented by glutamine, and the decline in its concentration could have pervading consequences for protein synthesis and final grain composition and quality [56,196].

#### 3.2.3. Agronomic Management Effects

The purpose of this section was to discuss the current agronomic management practices adopted within durum wheat production systems and explore how they can lead to improvements or losses in quality aspects.

The main topics covered in the reviewed articles included:Cropping systems;Crop sequences;Prevalent management practices (sowing date, tillage, and fertilisation).

Concerning **cropping systems**, some studies compared the effects of **conventional versus organic farming practices** on different quality attributes of durum wheat grains and semolina pasta.

Interesting results were obtained by Quaranta et al. 2010 [77] in three years of trials for six durum wheat varieties grown under conventional and organic cropping systems in different Italian regions. According to them, the organic yields were significantly lower (−16%) than those obtained in conventional cropping, excluding the southern locations where the difference was lower (−5%). Similar data were obtained for grain protein, even if the gap between conventional and organic systems was less than 1.0% [3]. Regarding the levels of deoxynivalenol (DON) contamination, the organic cropping system instead allowed for good results not only in Southern Italy, where fungal infections were limited but even in the locations of Central Italy, which are more exposed to the risks of hazardous fungal pathogens attacks. This fact was mainly attributed to the rotations with non-host *Fusarium* crops such as leguminous. As a result, Quaranta et al. 2010 [77] proved the importance of both southern vocational areas for durum wheat farming and sound agronomic practices for the control of *Fusarium* and associated mycotoxins contamination.

Comparably, Fagnano et al. 2012 [78] found that organic production determined a reduction in protein content, gluten content, and quality, thus stressing the core problem for the organic sector, which is reduced protein accumulation. Despite this, one of the varieties under study achieved the highest quality level in both conventional and organic cropping systems, thereby confirming that the choice of an adaptable genotype can allow for gaining satisfactory outcomes even under the organic regime. In addition, the authors recommended also rotating durum wheat with legume crops to ensure the necessary N is available at sustainable costs.

Further concerns have been raised regarding the use of nonselective and broad-spectrum herbicides, such as glyphosate, in conventional farming systems. Indeed, glyphosate and aminomethylphosphonic acid (AMPA) residues are often detected on soil, water bodies, and food products, causing detrimental effects on environmental and human health [197].

However, there was debate about whether the use of glyphosate should be banned, restricted, or encouraged as a result of the frequently contradictory opinions of scientists, regulatory organisations, the general public, and the different regulations adopted on a global scale. For instance, the International Agency for Research on Cancer (IARC) in 2015 reclassified glyphosate as a Category 2A (probable carcinogen), whereas the Joint Food and Agriculture Organization, the European Food Safety Authority, and the European Chemicals Agency kept on confirming glyphosate as not posing a genotoxic or carcinogenic risk to humans [198]. As a consequence, although the Codex Alimentarius allowed a residues limit of 30 ppm of glyphosate for wheat, some countries have lowered the permitted limits [24]. In Italy, for example, the pasta industry accepts a limit of under 10 parts per billion to meet the increasing consumers’ requests for low or zero chemical residues [24].

Regarding the effects of glyphosate application on durum wheat quality, Manthey et al. 2004 [98] investigated the impact of glyphosate on wheat quality traits and concluded that its pre-harvest application could decrease hectolitre weight, thousand kernel weight, and kernel size. Other studies analysed the effect of glyphosate application on the chemistry of gluten proteins and shikimic acid accumulation [99] and on starch chemistry [100]. The results from Malalgoda et al. 2020 [99] indicated that pre-harvest glyphosate spraying decreases the molecular weight of SDS extractable and unextractable proteins whilst increasing the amount of shikimic acid accumulation. Thus, [99] advocated for investigating the changes related to the pre-harvest glyphosate spraying to provide insights on how correctly apply it without causing any related negative effects on the functionality of proteins. Similarly, Malalgoda et al. 2020 [100] studied the effect of glyphosate spraying timing (recommended stage vs. early stage) on wheat starch chemistry and discovered that the starch structural properties and thermal behavior were altered, especially if applied at the soft dough stage.

The type of cropping systems also influences the health-promoting properties of durum wheat: these evaluations were carried out in three studies among those reviewed, Nocente et al. 2019 [76], Fares et al. 2019 [108], and Pandino 2020 [75].

Nocente et al. 2019 [76] investigated the effect of the cropping system (conventional vs. organic) and tillage system (conventional vs. reduced tillage) on kernel bioactive compounds of durum wheat in a two-year study. Their results stressed the importance of growing year and management: indeed, high rainfall in the first year of the experiment was likely responsible for the loss of soil organic matter, annulling the differences between the two agronomic management systems; conversely, when rainfall was regularly distributed, the differences were substantial, with higher alkylresorcinols and total phenols contents in organic than in conventional farming. With regard to tillage intensity, alkylresorcinols and total phenols were positively affected by deeper tillage in organic management, whereas yellow pigments and total antioxidant activity were favored by reduced tillage in conventional farming.

The effects of organic and conventional cultivation under equivalent nitrogen fertilisation (as organic N and mineral N for organic and conventional cultivation) on antioxidant content and composition and quality traits were assessed by Fares et al. 2019 [108] using several old genotypes of three different kinds of wheat (*T. dicoccum, T. durum, T. spelta*). Compared with conventional farming, organic farming positively contributed to test weight, whilst other quality traits, such as protein and gluten contents and sodium dodecyl sulphate (SDS) sedimentation volume, were 19.2%, 9.3%, and 22.7% lower, respectively. Despite the fact that the protein content was lower under the organic regime, it was still high: thus, with adequate organic nitrogen fertilisation, the technological quality traits of organic flours can be sustained. At the same time, although free and bound phenolic acids did not significantly differ, total phenolic content (TPC) was significantly higher for organic versus conventional farming.

Coherent results were observed by Pandino 2020 [75] in examining the effects of cropping systems on the level of some nutrients, phytochemical compounds, and antioxidant activity, as well as on the yield and the rheological indices. According to them, the conventional cropping system appeared to determine better performance than the organic ones in terms of grain yield, protein content, alveograph dough strength (W), and wet gluten and microelements levels (Cu, Fe, Mn, Zn). On the contrary, total phenols content and antioxidant activity were highest in the organic cropping system. This could help to choose genotypes able to provide higher daily intakes of minerals and health-promoting compounds.

Concerning **tillage practices**, a renewed interest in the effects of the **transition from conventional to conservation management** was found in the reviewed studies. This was due to some concerns that needed to be addressed when conservative agriculture was first introduced, such as the possible diffusion of durum wheat diseases, unstable yields, and grains’ technological quality [199].

Specifically, Calzarano 2018 [79] looked at the effect of different patterns of soil treatments and crop sequences (conventional vs. zero tillage; monocropping vs. crop rotation with faba bean) on productivity, grain quality traits as well as on disease incidence and severity in durum wheat. The results of two years of data from a long-term experiment revealed that the conservation agriculture techniques (zero tillage + crop rotation) improved the grain yields, protein accumulation, gluten quality, and colour appearance. The highest grain yields associated with zero tillage and durum wheat–faba bean rotation were also explained by Pagnani 2019 [80], who observed an increased number of ears m^−2^, and a higher N-remobilisation to the grains.

Regarding tillage practices, another interesting study comes from Ruiz 2019 [81], who made the assumption that conservation agriculture demands varieties capable of sustaining good yields with few inputs in the face of unpredictable climatic conditions. Therefore, this study analysed the agronomic and morphological characteristics, as well as technical quality features of several bread and durum wheat genotypes, including landraces, intermediate and modern varieties, grown under minimum tillage, and no-tillage. According to their findings, a positive influence of no-tillage on traits related to grain and biomass yield was evidenced, whereas the tillage system and its interactions with year and with genotype did not have any relevant effect on quality traits. This makes it possible to assume that the quality of any variety will be unaffected, whatever the conservation tillage practice is used [81].

Regarding the effects of **fertilisation practices**, it should be noted that the latter accounted for the majority of studies dealing with agronomic management factors. This was not surprising to the authors’ opinion since researchers have always been interested in the modification of the quality characteristics of wheat grains influenced by basic fertilisation and are recently more and more debating on how to improve the cost-effectiveness and environmental sustainability of such fertilisation supply [194,200,201,202].

The reviewed studies particularly investigated different aspects inherent to the use of fertilisers, such as the nutrient, rates, times, and method of application, and with regard to diverse quality aspects.

**N fertilisation** is a common practice adopted to increase yield and grain protein content. The need for increased protein content is due to the extensive use of continuous durum wheat and durum wheat-fallow systems, which tend to decrease this content below 13%, the minimum level requires for pasta-making. Moreover, high-yielding modern cultivars have prompted an increase in the use of N fertilisers since they need a higher fertilisation rate than old ones to reach their maximum grain yield and protein content [6].

Among the reviewed studies, the first paper to deal with N fertilisation effects on grain yield and grain protein content was carried out by Colecchia et al. 2013 [86], who experimented with different N application rates and timing on six durum wheat cultivars. Their results showed that N application at the sowing date did not impact the cultivars’ productivity and quality, suggesting the possibility of postponing the fertilisation and using it only during the critical phase for durum wheat, that in semiarid conditions coincides with the end of tillering and beginning of stem elongation. At the same time, higher N fertilisation levels did not cause any meaningful increase in yield and protein because it is likely that the excess N is leached during the period of higher rainfall, causing environmental and economic losses.

Morari et al. 2018 [88] evaluated the effects of different N fertilisation rates on protein quantity and also quality. In that case, higher N fertilisation levels were found to induce high protein contents and improve gluten proteins, as well as technical quality, but this practice was discouraged due to its possible contribution to the soil, air, and water pollution risks. Moreover, as stated by Marinaccio et al. 2016 [92], the effectiveness of N fertilisation on protein content and other quality traits depended also on soil texture and rainfall during growing seasons. Therefore the management of N fertiliser should be carried out also considering these interactions.

Another interesting study from Pasqualone et al. 2014 [203] evaluated the effects of the use of **composted sewage sludge**, as it is and in combination with common mineral fertilisers, on durum wheat productivity, technological quality, and antioxidant properties. The authors evidenced that sewage sludge application greatly increased the productivity as well as the concentration of phenolic compounds, antioxidant activity, and technological quality of durum wheat whole flours. With regard to phenolics and antioxidant activity, a further increase can be achieved by employing a combination of composted sewage sludge and mineral fertilisation.

One more approach of durum wheat nutrition management, not conflicting with environmental protection, deals with the application of **plant biostimulants**, able to modulate several functions of the plant, such as nutrient efficiency, abiotic stress tolerance, humification, and many others. Moreover, Pichereaux et al. 2019 [96] studied and reported interesting results about the influence of marine (DPI4913) and a fungal (AF086) biostimulants applied to leaves on grain yield, protein concentration, and composition of durum wheat grains. Indeed, both types of biostimulants induced an increase in grain yield and protein quantity, as well as a modification of the protein composition. Specifically, the proteome analysis allowed the authors to identify 50 proteins that were differentially represented after the treatments, all of them involved in metabolic reactions and processes, including storage, regulation processes, and defense response against abiotic and biotic stresses.

Similarly, Pačuta 2021 [95] studied the impacts of brown seaweed-based and humic substance-based biostimulants on yield and grain quality. The biofertilisation was realised three times during the vegetation for both substances, and it has increased the grain yield compared with the non-fertilised variant. However, the best results were obtained on the humic acid variant. Moreover, the study found no evidence of a statistically significant effect of biofertilisation on the quality traits (protein and gluten content, vitreousness, falling number, bulk density), even if the latter always maintained or slightly exceeded good levels.

Lastly, it is undiscovered that good effects on nitrogen supply and durum wheat quality can be obtained by adopting agroecological practices, such as **crop rotation and intercropping, especially with legume crops**. Indeed, besides the advantages of using leguminous species as a fundamental tool for the maintenance and management of soil fertility, intercropping durum wheat with grain legumes is known to increase the protein content in the durum wheat grains, so it could be used to enhance the quality [101,102,204].

Some studies were instead focused on fertilisation and biofortification for **some critical micro-nutrients** that are currently present in limited concentrations in the soils or that are difficult to be accumulated in the grain, thus leading to malnutrition.

For instance, **Sulphur (S)** deficiency in agricultural soils has become widespread in many European countries, and S fertilisation in the study of Ercoli et al. 2011 [93] showed positive effects on grain quality and pasta quality for all measured rheological and technological features, such as dry gluten, SDS volume, and alveograph indices.

Similarly, Abdoli et al. 2016 [84] addressed another important micronutrient deficiency, the one related to **Zinc (Zn)** in common and durum wheat growing in calcareous soils. According to this study, both soil and foliar zinc application methods could improve yield and grain Zn concentration. However, bread wheat generally had better agronomic traits; grain yield; and Zn, Fe, Cu, and Mn concentrations in grain compared with durum wheat. In addition, different Zn treatment methods also decreased the phytate content, the latter being acknowledged to be chelating of Zn. For these reasons, the use of zinc fertilisers appeared to be a reasonable temporary solution to alleviate zinc deficiency. With regard to the latter problem, another team of authors, namely Tran et al. 2019 [90], pointed out that the bioavailability of Zn, as well as that of **iron (Fe)** in the grain of cereals, can be overestimated by failing to consider the abundance of phytic acid. In this work, the authors specifically looked at the growth and nutritional responses of durum wheat to arbuscular mycorrhizal fungi and to diverse soil Zn concentrations. It was observed that the inoculation increased the concentration of phytic acid in durum wheat grain whilst having no effect on the concentration of Zn and Fe: this consequently reduced the predicted bioavailability of grain Zn and Fe, which could result in a decrease in the nutritional quality of the grain.

Likewise, Melash et al. 2019 [94] conducted a field experiment to evaluate the effect of foliar micronutrient application on grain yield and quality traits of durum wheat, and in line with other authors, observed that the application of ZnSO_4_ and FeSO_4_ has increased the concentrations of both elements in the grains, although with differences among genotypes. Thus, they confirmed that agronomic biofortification could be a strategy to overcome health issues related to Zn and Fe deficiency if applied to varieties with well-known genetic responses to such treatment.

The other two studies investigated the use of **Selenium** (Se) to improve durum wheat agronomic performances and nutrient uptake and the quality of the derived food products [87,91]. De Vita et al. 2017 [87] carried out two field experiments over three growing seasons to evaluate Se foliar fertilisation at various rates and timing of application on four durum wheat varieties for pasta-making. Their findings confirmed the effectiveness of foliar Se fertilisation in increasing Se concentrations and highlighted more significant responses to the treatment in landraces and old varieties compared with the modern ones. Moreover, the increased Se content in the grains did not modify grain quality features, as well as the sensory features of the final pasta. Similarly, Ayed et al. 2022 [91] explored the effect of Se foliar supply on grain yield and technological quality traits of different durum wheat genotypes, including advanced lines and modern varieties grown under semi-arid and sub-humid regions in Tunisia. As for De Vita et al. 2017 [87], Se responses were genotype-specific both within and among the two studied areas, and Se foliar application was confirmed to be a safe way of increasing grain yield and quality.

### 3.3. Technological Factors

The overall quality of durum wheat pasta also depends on many technological factors, dealing with the main processing operations that are carried out to transform grains and semolina. Hence, it is to remark that each kind of food technology, according to its variables, such as working principles, time, temperature, and level of intensity, can determine both enhancements or detriments of the nutritional and sensory properties of the final product. With regard to the pasta sector, scientific research is constantly evolving, both with studies on the improvement of the raw material and on the innovation of processing plants [205]. As stated by Hidalgo et al. 2010 [130], such effort is needed to identify the best strategies to control the degradation of nutrients and bioactive compounds during the manufacturing stages, thus prompting the production of foods with improved nutritional properties.

Concerning pre-milling or pre-processing procedures, **debranning** was investigated among different studies for the production of durum wheat ingredients and/or pasta with higher amounts of health-relevant components and structural features giving good sensory properties [119,120,121]. Indeed, by gradually removing the kernel layers through an abrasive scouring, debranning has been shown to:-Increase semolina yield and quality;-Reduce the ash content;-Reduce the contamination by mycotoxins, agrochemicals, and heavy metals.

In that way, it provides the possibility of collecting fractions with selected content of bioactive compounds, which can be used as supplements for the formulation of functional foods [206]. About this, Martini et al. 2015 [120] applied a twelve-step sequential debranning process to durum wheat and monitored the content of phenolic acids and total antioxidant capacity in both the bran fractions and the debranned kernels. From the results of their study, it can be inferred that: (i) total phenolic acids content was much higher in the bran fractions than in debranned kernels in all samples collected during the debranning, confirming that these compounds are mainly accumulated in the external layers; (ii) the different phenolic acids forms showed different occurrence among the bran fractions, suggesting the possibility to find the optimum debranning level needed to produce fractions, and debranned kernels with the highest contents (iii) the total antioxidant capacity assessed in the bran fractions and debranned kernels were higher than displayed by wholemeal, semolina and coarse bran, produced from the same durum wheat employed in the study by traditional milling. Therefore, the debranning process was proposed as a valuable approach for obtaining: bran fractions, which can be added to the formulations of wheat-based foods with enhanced nutritional and health value, and debranned kernels to be entirely milled for the production of less refined and safer products.

Similar conclusions were also reported by Ficco et al. 2020 [119], whose study investigated the application of the debranning process to purple durum wheat to identify the exact conditions at which such technology provides the flours with the best characteristics, that is to say, higher contents in bioactive compounds, and essential minerals, and lowered amounts of toxic trace elements, such as Pb, Cd, Hg, As, Sn, Ni, Sr, V, and Cr. In their study, purple durum wheat genotypes were chosen for their richness in anthocyanin pigments and antioxidant properties [207], and six debranning times were applied (from 30 to 180 s). The latter generated six corresponding debranned grains (DG-1 to DG-6): of these, DG-1 and DG-2 seem to provide the optimal trade-offs by containing high levels of antioxidant compounds and essential elements (90% and 89.5%, respectively) and reduced amounts of the toxic elements (loss of 11% and 24.5%, respectively).

Furthermore, Ciccoritti et al. 2017 [121] integrated the results of Martini et al. 2015 [120] by exploring the use of debranning products for pasta production and checking the content of nutrients, bioactive compounds, and antioxidant activity throughout the entire process. Specifically, the two pasta samples, one produced with bran fractions-enriched semolina (BF pasta) or the other with micronised debranned kernels (DK pasta), displayed significantly higher content of phenolic compounds and dietary fibre than the control pasta (made with traditional semolina). Moreover, minimal effects on sensory properties were noticed by the authors.

**Micronization**, as mentioned previously, is another unconventional procedure. It entails an ultra-fine grinding procedure that reduces the particle size of matrices, including bran fractions and kernels, enhancing the bioaccessibility of various compounds, including phenolic acids, the solubility of dietary fibre, and the bioavailability of digestible carbohydrates [124]. With regard to this point, when Martini et al. 2018 [97] compared pasta produced with semolina and pasta obtained with micronised kernels, they discovered that the conventional method had a detrimental impact on both the overall antioxidant capacity and the concentration of phenolic acids, particularly during the milling process, as it is related with the removal of the outer layers of the kernels. In contrast, the authors were able to produce whole-wheat pasta that retained the number of antioxidant components even when cooked, thanks to the micronisation process.

Finally, Ciccoritti et al. 2019 [123] proposed a pre-milling mild debranning, followed by micronisation and air classification to produce high-value fractions to be added to traditional semolina for the manufacturing of naturally enriched pasta. **Air-classification** was adopted to transport the micronised samples towards a series of ascending airflows which separated heavier gross and fine particles, allowing their collection. This novel technological process permitted to produce of enriched pasta that increased up to 53% in soluble polyphenols, 121% in alkylresorcinols, 64% in arabinoxylans, 113% in dietary fibre, and 20% in resistant starch. In addition, the pasta showed good cooking quality and appreciable sensory features.

Concerning traditional primary processing, **milling** is a complex procedure of progressive grinding and sieving aimed at breakup wheat kernels and separating the endosperm from the bran, preserving grain quality. As the outer parts of the kernel, especially the bran, the aleurone layer, and the germ, are richer in bioactive compounds and minerals when compared with the starchy endosperm, conventional milling reduces their content in semolina and concentrates them in the milling by-products [125]. However, lipid peroxidation and rancidity process are known to occur in stored wholewheat flour and negatively affect its quality, shelf-life, and end-use.

The effect of milling on durum wheat on starch, dietary fibre, mineral elements, phenolics, and antioxidant activity was assessed within diverse studies among those reviewed [117,125,126,127]. This information was crucial to evaluate the significance of nutrient and micronutrient losses as a consequence of conventional milling and to provide a basic understanding to establish, for instance, whether fortification is necessary, which element should be supplemented (in a bioaccessible form), and at what levels [125].

For instance, Di Benedetto et al. 2013 [126] found that the milling process determined a negative effect on ABTS·+ [2,2′-azinobis(3-ethylbenzothiazoline-6-sulfonic acid)] scavenging activity of semolina and pasta compared with the whole meal, for both hydrophilic and lipophilic extracts. This result may be attributed to the removal of the outside layers of the kernel, including the pericarp, a part of the aleurone layer, and germ, which is richer in functional components, and responsible for antioxidant activity.

Similarly, Di Silvestro 2014 [117] evaluated the effects of two milling systems (water and stone) differing in the temperature generated during grinding (namely, 30 and 60 °C) and of storage conditions on the concentration of phytochemicals. By comparing the milling techniques, the stoneground wheat grains (60 °C) showed the highest amylose and resistant starch amounts, which could be advantageous for lowering the glycaemic index of the final products. In contrast, the polyphenol content was significantly reduced in the analysis of flour samples produced using the stone mill compared with the water mill.

Cubadda et al. 2009 [125] focused their research on the effect of processing on the contents of eight minerals—i.e., calcium, copper, iron, magnesium, phosphorous, potassium, selenium, and zinc. Their results showed that conventional roller milling significantly reduced the content of each mineral in durum wheat grains. Specifically, the concentration losses widely changed based on the element, according to the following retention order: Se > Ca > Cu > P ≈ K > Fe > Mg ≈ Zn, from 16 % for Se to 66% for Zn. In addition, calcium increased after cooking, whereas the concentrations of the other minerals were either unchanged or slightly reduced.

Some studies, among those reviewed, instead focused on the **storage conditions** of both grains and semolina before the pasta-making process. Indeed, the quality of grains and semolina need to be preserved and maintained through appropriate temperature and moisture conditions, as well as protective treatments, respectively, before and after milling. During storage, indeed, the grain quality can be deteriorated because of both abiotic and biotic factors, and prolonged storage can cause adverse effects on processing and end-product features.

Variations in abiotic factors, such as the duration of the storage, the temperature, and the moisture of grain and semolina, caused changes in thiol (SH), lipids, carotenoids, and phenols contents, with effects on the technological and nutritional properties [116,117,118,208]. For instance, according to Tomić et al. 2013 [116], the amount of free sulphydryl groups increased during postharvest wheat and flour maturation, as well as with the increase in temperature and gluten incubation time. However, the most unstable components, especially in whole-wheat flours, are the lipids that are subjected to enzymatic degradation reactions, leading to a reduction in functionality, palatability, and nutritional properties [209]. Concerning carotenoids, Mellado-Ortega et al. 2015 [118] found that under long-term storage conditions, the total carotenoid content decrease according to a temperature-dependent first-order degradative kinetic model. Moreover, according to Di Silvestro 2014 [117], after 6-month storage of flour at a temperature ranging from 20 to 25 °C, a decrease in soluble dietary fibre and bound polyphenols was observed, whilst other wheat grain components, such as starch, remained unvaried.

Regarding biotic factors, instead, pests consisting of insects and mites need to be controlled at various stages such as on-farm storage, bulk storage, and during transport [210]. Fumigants are especially used to control pests and insects in food commodities to reduce product losses. However, such chemicals are also associated with serious risks such as ozone depletion, insect resistance, and residues on the grain surface, and therefore alternatives are constantly searched [210]. Regarding these aspects, Hwaidi et al. 2016 [114] assessed the effect of fumigation of grain and commercial semolina with sulfuryl fluoride (SF) under different conditions on grain germination and the technological quality of the grain, semolina, and pasta quality. SF appreciably reduced the germination percentage of fumigated durum wheat, with increasing impact under higher fumigant concentration, whilst it had little to no effect on grain test weight, thousand kernels weight, hardness, protein content, semolina ash content, and mixograph properties. At the highest SF concentration (31.25 mg/L for 48 h), there was a tendency for pasta cooking loss to be increased but still acceptable, and other pasta properties were largely unaffected.

Another study, instead, focused on the effects of gaseous ozone treatments on DON, microbial contaminants, and technological features of wheat and semolina [115]. Ozone is a powerful disinfectant, which readily degrades specific contaminants, including several mycotoxins, and is a promising fumigant against insects and a wide range of microorganisms that can survive and grow in the grains during storage [211]. However, due to its oxidant-reduction potential, ozone can react with many functional groups present in lipids, proteins, and carbohydrates, affecting rheological properties and technological acceptance of flours and doughs [115]. Specifically, results from Piemontese et al. 2018 [115] reported the efficacy of gaseous ozone treatments in reducing DON, DON-3-Glc, bacteria, fungi, and yeasts in naturally contaminated durum wheat and allowed to identify the optimal conditions, which do not affect chemical and rheological properties of durum wheat, semolina, and pasta (55 g O_3_ h^−1^ for 6 h).

Within the secondary processing, the pasta production process involves **mixing and kneading**, **extrusion, and drying** to reach about 12.5% of moisture and then **cooking**. For “fresh pasta”, it should be underscored, however, that drying is replaced with pasteurisation with the aim of stabilising the previously extruded product.

To date, the most important technological advances in pasta processing concerned:-The adoption of different types of mixers, mixing speeds, times, and pressures;-The introduction of die inserts lined with Teflon, partially replacing bronze ones;-The increase in drying temperatures from 75 °C to 100 °C and even above, thus reducing the drying time from the original 48 h to only 2–3 h [143].

However, the pasta manufacturing process can be thought of as “mature technology” due to its wide-reaching diffusion and the few innovations applied to this process in the last years [212].

The initial step of the pasta manufacturing process consists of **mixing** semolina and water: mixing times, speeds, and pressures; hydration level; and water temperature vary based on the processing conditions, the moisture content, the particle size of semolina, and the final pasta shape. In particular, **mixing and kneading under vacuum** conditions have been recently designed for pasta and noodle manufacturing applications, as it allows for the following:-Uniforming the hydration process and creating a more homogenous water–flour mixture;-Avoiding the incorporation of oxygen, thus eliminating air bubbles in the final products and limiting enzymatic activity responsible for the antioxidant degradation [140,213,214,215,216].

In this context, several studies such as Hidalgo and Brandolini 2010 [128], Hidalgo et al. 2010 [130], and Fratianni et al. 2012 [140], as reviewed in this paper, investigated the effects of processing stages on tocol and carotenoid contents of different wheat-based products and confirmed that for pasta the degradation of such antioxidants is mostly limited to the mixing and kneading steps. In addition, the flour particle size, depending on the genotype, the grain hardness, and the milling procedure, is another possible source of colour variation [141].

Hidalgo et al. 2010 [130] pointed out that the kneading led to a higher total carotenoids degradation in pasta compared with bread and biscuits due to the longer time of the phase (7 vs. 3 min) and found significant differences in the percentage of degradation based on the wheat species used for the pasta formulation (54% in bread wheat, 34% in semolina and 15% in einkorn). This was mostly attributed to the fact that the wheat species under study have diverse lipoxygenase activity (bread wheat > durum wheat > einkorn).

Similar findings were obtained for tocols by Hidalgo and Brandolini 2010 [128], who observed higher losses during the kneading phase (44.2%), and the no-vacuum extrusion step (29.7%). As for Hidalgo et al. 2010 [130], Hidalgo and Brandolini 2010 [128] results agreed on identifying einkorn as a better provider of tocols, in the end, products than durum and bread wheat.

In line with them, Hidalgo and Brandolini 2012 [129] stressed the need to select genotypes with high yellow pigment colour and low enzymatic activity, to maintain higher antioxidant features and a better appearance in the final products. Utilising species/subspecies/accessions differing in their enzymatic activity or their esterification ability could be useful for producing foods with improved technological and nutritional quality features [43,131]. However, other approaches to increase carotenoids and tocols stability were also proposed, such as the enrichment of pasta with encapsulated carrot waste extracts in oil [177]. 

Regarding **extrusion**, **bronze dies** have been partially replaced by **Teflon inserts**, especially in high-speed plants, as they are associated with some disadvantages, such as:-Faster wear of the die insert;-Lower production yield of the press;-Lower breaking strength;-Higher susceptibility to insect attacks [132,133].

Despite this, bronze inserts give a rough surface to pasta, which better binds the sauce, with greater appreciation by consumers. This is one of the primary motivations for which several pasta companies still prefer to use bronze inserts, along with the valorisation of traditional raw material (for instance, semolina of old varieties and landraces) and the adoption of low drying temperatures. Indeed, such pasta production models tend to appear appealing to a niche of consumers who value authenticity and tradition.

Valuable results about the influence of extrusion on pasta quality feature other than surface characteristics were reported by several studies that were reviewed in this paper, namely Hidalgo and Brandolini 2010 [128], Hidalgo et al. 2010 [130], Abecassis et al. 1994 [134], Debbouz and Doetkott 1996 [135], Zardetto and Rosa 2009 [136], and Pagani et al. 1989 [137].

For instance, Abecassis et al. 1994 [134] found that an increase in the extrusion temperature (from 35 °C to 70 °C) is able to cause an increase in cooking losses up to 250%. However, the damage due to high extrusion temperature could be mitigated by increasing the level of hydration and the rotation speed of the screw. In this regard, Debbouz and Doetkott 1996 [135] proposed extrusion temperatures between 45 and 50 °C and hydration levels of 31.5–32% as optimal conditions to reduce pasta cooking losses.

Moreover, attention should be posed to the use of vacuum: as in the case of mixing and kneading, applying vacuum to the extrusion or lamination phases significantly reduced carotenoids and tocols degradation [128,130,136]. Additionally, another function of a vacuum for the lamination process could be to give extensibility and consistency similar to those obtained with the extrusion [136].

Along with mixing and extrusion, the **drying** process plays a key role in ensuring the quality of pasta since its conditions may cause thermal and mechanical damage, with negative consequences on the texture and the sensory properties of the product [35]. **High-temperature drying (HT)** advantages consist of a reduction in microbial contamination and improvement of the cooking quality of pasta, whilst disadvantages concern the occurrence of Maillard-type reactions that leads to protein modifications due to cross-linking with other compounds, which can affect both protein digestibility and antigenicity [138,143]. The extent of the Maillard-type reactions in pasta dried at high temperatures can be especially determined by measuring the furosine level and is significantly related and dependent on the α-amylase activity and the amount of reducing sugars in the semolina [139].

Within this context, the study by De Zorzi et al. 2007 [143] was enlightening since it evaluated the **effects of different drying temperatures** on the digestibility and potential allergenicity of pasta samples cooked in boiling water. Temperatures considered by the study were both in the range of those used by industry (60, 85, and 110 °C), of a traditional room temperature drying (about 20 °C), and an ultra-high temperature treatment (180 °C). This latter temperature, which is rarely used in the production of industrial pasta, was considered to evaluate the effects of very drastic heating. The findings confirmed that protein digestibility is impaired by drying temperatures, and those semolina proteins aggregated with each other during drying, forming strong and stable interactions. Specifically, for the samples treated at temperatures in the range of those by industry, this aggregation was due to the formation of disulfide bonds and hydrophobic interactions, as almost complete solubilisation was observed by using a solvent able to disrupt these interactions. In contrast, this was not possible to be detected on the proteins of the samples dried at 180 °C, which were characterised by the presence of aggregates stabilised by irreversible interactions. Moreover, IgE-immunoblotting and IgE dot blotting were carried out using the sera of patients with food allergies to wheat in order to detect the potential allergens coming from the cooked pasta samples before and after digestion. The latter analysis revealed that, regardless of the degree of heat treatment, the digesting process was unable to entirely eliminate the presence of IgE-reactive peptides, even if it was adequate to break down the proteins in the samples dried up to 110 °C.

Besides protein digestibility, another aspect to be evaluated is the effect of drying temperatures and cooking on starch digestibility, as its degradation is affected by both the physical modifications (mainly gelatinisation of starch) and the compact protein matrix surrounding its granules. The changes in starch structure during pasta manufacture can be tailored to produce slowly digestible starch for pasta with a reduced glycemic index.

According to Petitot et al. 2009 [145] and Petitot and Micard 2010 [146], changes in dried and cooked pasta structure due to the drying process were moderate up to 70 °C and increased at higher temperatures, especially for very high temperatures (90 °C) applied at the end of the drying profile. In fact, the latter drying process resulted in a 10% reduction in protein digestibility and induced a lower starch digestibility in cooked pasta.

Finally, Stuknytė et al. 2014 [144] adopted a new static protocol for in vitro digestion [217] that allows assessing the digestibility of cooked spaghetti dried under low temperature (LT) or HT conditions by measuring the release of amino acids and sugars in the digestates. Following their results, intense starch degradation was observed at the intestinal level, as demonstrated by the release of 12.6 and 15.9 g 100 g^−1^ of sugars in the digestates of LT and HT-cooked spaghetti, respectively.

Therefore, contrary to Petitot et al. 2009 [145], Stuknytė et al. 2014 [144] stated that HT drying does not decrease the in vitro digestibility of starch, despite the slightly firmer protein network. Instead, in the same samples, diverse amounts (16.3 and 12.5 g 100 g^−1^ protein) of free amino acids were found, thus indicating that the HT sample was slower to digest.

### 3.4. Pasta Formulation Factors

Care is recommended to be taken to produce very high-quality pasta, starting with the choice of the raw materials, ingredients, and additions, moving on to the processing line parameters, and finishing with the packaging requirements [218].

Recent advancements in the pasta sector, in particular, were mainly focused on ways to improve the nutritional value of the product and to provide potential health advantages to consumers through the use of unconventional ingredients [218,219]. Among the latter, this paper’s team of authors identified and classified the papers dealing with changes in pasta formulation based on the ingredient added, as follows:High-fibre ingredients (beta-glucans, inulin, resistant starch, bran fractions);High-protein ingredients (protein concentrates, legume flours, insects);Probiotics (*Bacillus coagulans* GBI-30);Other highly nutritious and functional ingredients.

For the first class (**high fibre ingredients**), the literature demonstrated that adding dietary fibre to pasta, particularly its soluble fraction, causes a decrease in the glycaemic index of the products [149]. In addition, high-fibre grain products are also usually rich in minerals, vitamins, and biologically active compounds, all of which offer additional health advantages [149]. However, due to the presence of bran, whole-grain foods generally have inferior technological and sensory properties than refined ones [121]. For instance, according to Padalino et al. 2015 [220], wholemeal spaghetti samples showed an improvement in the chemical composition by preserving high protein and insoluble dietary fibre contents, but they had worse cooking quality features, as demonstrated by high cooking loss, low elasticity, firmness, and colour, compared with the semolina spaghetti.

To overcome these drawbacks typical of wholemeal products, Sobota et al. [111] suggested using a common **wheat bran** additive of up to 30% and favoring wheat bran with fine and uniform granulation and possibly high content of gluten.

With regard to granulation, Alzuwaid et al. 2020 [147] examined the impact of the incorporation at different levels of several bran fractions with diverse particle sizes on the technological and health-promoting properties of pasta. Their findings showed that higher levels of bran incorporation (especially at 20%) decreased pasta quality, but authors highlighted that such negative impact could be reduced using finer bran. Moreover, the study found that the increase in antioxidants (by up to 65%), ferulic acid (up to 400%), and phytosterols (up to 130%) due to the incorporation of bran was insensitive to the particle size above 10% incorporation. Thus, it was recommended to make use of finer fractions, especially when the bran is added to pasta at a 20% level, since they can provide good quality performances as well as higher phytochemical contents.

From a sensory quality point of view, also the results from Steglich et al. 2015 [148] stressed that consumers tend to like pasta more when a decreasing particle size of semolina is used in its formulation, probably because of the smoother pasta surface and milder whole wheat aroma.

Similar ingredients are represented by **resistant starch, inulin, β-glucans**, as well as **non-conventional flours** such as psyllium, oat, and sorghum flours.

Specifically, Aribas et al. 2020 [152] reported significant advantages of the supplementation of pasta with **resistant starch type 4 (RS4)** compared with bran supplementation. Pasta formulated with such ingredients had a better look, texture, and sensory qualities, reduced cooking loss, higher firmness, higher total dietary fibre, improved mineral bioavailability, and lower in vitro glycaemic index than pasta supplemented with wheat bran.

Chillo et al. 2011 [153] instead investigated the in vitro glycaemic impact and cooking quality of spaghetti made with semolina enriched with one of either two **types of β-glucan barley concentrates**, namely Glucagel^®®^ (GG) and Barley Balance™ (BB). The addition of up to 10 g/100 g of both types of β-glucan did not significantly alter the pasta cooking quality compared with the control and improved the fibre-associated nutritive properties of the spaghetti. Moreover, BB showed a better capacity to lower the glycaemic index of the pasta compared with GG.

Moreover, **inulin** is acknowledged as an ingredient that promotes health through several mechanisms, dealing with counteracting constipation, supporting the growth of microflora in the digestive tract, and managing blood sugar levels. Therefore, because the interactions between starch and inulin could further slow starch digestion and thus lower the glycaemic response, its addition to pasta was proposed to obtain a functional food [154]. In this respect, both Garbetta et al. 2020 [154] and Padalino et al. 2017 [155] focused on the effects of the addition of inulin with different polymerisation degrees on the chemical and sensory properties of spaghetti based on whole-meal durum wheat semolina. In particular, Garbetta et al. 2020 [154] provided specific data on how the polymerisation degree affected sensory quality, whilst it did not influence the glycaemic index values. Moreover, their results demonstrated that, for all the cultivars under study, the pasta sample produced with the cardoon roots inulin released a statistically larger amount of inulin in the digestive tract than the one produced with chicory, thus underscoring its importance prebiotic. In particular, Garbetta et al. 2020 [154] evidenced that the polymerisation degree affected most of the sensorial attributes of pasta, whilst it did not influence the glycaemic index values. Moreover, the results showed that for all the studied cultivars, the pasta sample produced with the high polymerisation-degree inulin (cardoon roots) released a statistically higher amount of inulin in the digestive tract than the one produced with the low polymerisation-degree inulin (chicory), thus highlighting its importance prebiotic function.

Padalino et al. 2017 [155] agreed with this by stating that the addition of cardoon roots inulin to the dough provided better results and suggested the selection of inulin with a high polymerisation degree for pasta enrichment.

Other studies regarded the addition of other types of flours, such as **psyllium, oat, and sorghum flours**, since they are naturally rich in dietary fibre [151,156]. A positive effect of the addition of different types of dietary fibre in pasta on in vitro digestion was well documented [151,156]. Therefore, at this stage, it would be interesting to have confirmation from in vivo starch digestion analysis [151]. Moreover, additional research would be necessary to evaluate the acceptance of enriched pasta from consumers [151].

Incorporating **high-protein ingredients** into pasta is thought to be crucial for increasing the protein content, which is typically around 15%, and making up for durum wheat’s comparatively low quantities of the key amino acid lysine. Pasta with a higher protein content typically has better resistance to overcooking, high firmness, low stickiness, and low cooking loss [161]. Additionally, as protein content rises, starch digestion becomes less extensive due to a larger, more cohesive network that restricts access to amylase and lessens starch gelatinisation [161]. In terms of protein quality, instead, raising the dough strength of durum wheat by increasing the quantity of glutenin and the ratio of HMW-GS/LMW-GS did not guarantee an increase in pasta firmness [187].

Alzuwaid et al. 2021 [157] investigated if bran protein concentrate could be added to spaghetti to improve its nutritional value without impairing its quality. Compared with durum-semolina spaghetti, the wheat bran protein concentrate boosted the total quantity of essential amino acids and raised the protein level by 90%. Additionally, the overall quality of the spaghetti was comparable to the control (100% durum wheat semolina) up to 5–10% levels of incorporation but began to degrade at 20%, even though it was still superior to commercial wholemeal spaghetti made with bran addition in terms of cooking loss and appearance. Therefore, according to Alzuwaid et al. 2021 [157], such a low-value waste stream could be a promising alternative source of protein and a better way to generate added value for the milling sector.

Recent research has assessed various choices for enhancing durum wheat pasta with high-protein sources other than durum wheat, such as flour from insects and legumes. These options were evaluated in terms of their technological, nutritional, and sensory qualities. For instance, some studies were focused on **faba bean enrichment** of durum wheat pasta, as faba bean has larger amounts of dietary fibre and proteins, the latter of which are naturally abundant in the essential amino acid lysine [159,160,221]. Therefore, a human feeding trial performed by Chan et al. 2019 [221] showed that substituting durum wheat semolina with 25% faba bean flour, starch isolate, protein isolate, and protein concentrate had positive impacts effects on post-prandial glycemia and satiety. However, despite its ability to improve nutritional value, faba bean addition is linked to changes in the rheological characteristics of pasta. In that regard, Greffeuille et al. 2015 [160] found that high-temperature drying strengthened the textural properties of 35% legume pasta and induced several benefits, such as a decrease in appetite and improved digestive comfort, besides having no effect on pasta’s low glycemic and insulin indices [159]. Similarly, another study evaluated the replacement of semolina with **lupin flour**, and, in this case, it was also found that the protein, dietary fibre, and mineral concentrations were increased [162]. Additionally, it was tested and recommended to combine lupin flour with RS4 to develop pasta with appropriate cooking quality, sensory qualities, and enhanced nutritional properties [162].

**Crickets and cricket powder**, although very different from other ingredients mentioned above, may also be considered valuable protein supplements in the production of wheat-based food products, as they contain considerable amounts of proteins, along with lipids (especially polyunsaturated fatty acids) and minerals [158]. Depending on the type of insect, its development stage, and the type of breeding, the protein content can considerably vary, from 13% to over 77% of dry matter [222].

The European legislation considers whole insects and their parts as novel foods (NF) with specific microbiological, chemical, and environmental risks to be monitored [223,224,225,226], whilst other countries do not yet have regulations regarding edible insects. Recently, the European Food Safety Authority (EFSA) Panel on Nutrition, Novel Food, and Food Allergens identified no other safety concerns than allergenicity for *A. domesticus* (AD) formulations and concluded that this novel food is safe under the proposed uses and use levels. The latter change depending on the formulations (frozen, dried, powder) and the food categories in which the ingredient is expected to be added (e.g., for dried pasta, the maximum use levels are 1 g NF/100 g and 3 g NF/100 g of AD dried and AD powder, and AD frozen, respectively) [224].

In Duda et al. 2019 [158], three levels of durum semolina replacement (5%, 10%, and 15%) with cricket powder were tested to examine the associated effects on the nutritional content, cooking and textural qualities, and colour, as well as consumer acceptance of the enriched pasta. According to the results, the quality traits of the enriched pasta significantly differed compared with those of the control pasta. In particular, the following aspects have been noted: a significant increase in the content of protein, lipids, and minerals; different cooking properties, such as a reduction in cooking weight and cooking loss, increased optimal cooking time, and higher firmness, and a general product’s colour shifting towards red and blue. Moreover, consumers involved in the sensory analysis accepted enriched pasta quite well.

Only two studies among those reviewed were found to deal with **probiotic pasta** [163,164]. This was expected since heat-treated foodstuffs, such as dry pasta, are not usually utilised as a probiotic carrier, as their processing includes a number of heat treatments to which most of the microorganisms cannot survive. Despite this, both Fares et al. 2015 [163] and Konuray and Erginkaya 2020 [164] concentrated their research on spore-forming probiotic microorganisms that have lately gained interest in the cereal industry for their ability to maintain their viability in heat-treated operations, such as boiling and baking. Both studies specifically analysed the inclusion in pasta formulation of the *B. coagulans* GBI-30 strain, which obtained the Generally Recognised As Safe (GRAS) status in 2012 from the U.S. Food and Drug Administration (FDA), following an assessment of toxicological studies [227].

Following Fares et al. 2015 [163], the probiotic strain remained viable during the pasta manufacturing and boiling processes, and its concentration in cooked samples (about 9.0 log CFU/100 g) would be considered sufficient to exert beneficial effects on the consumer. In line with this, Konuray and Erginkaya 2020 [164] analysed the probiotic pasta made with *B. coagulans* GBI-30 spores to establish its microbiological characteristics and quality parameters at 0 and 6 months of storage. Interestingly, the *B. coagulans* GBI-30 count was above 6 log CFU/g even after the pasta had been stored at room temperature for six months and cooked for eight minutes at 100 °C.

With regard to the last class **(other highly nutritious and functional ingredients)**, it was defined in that general way by this team of authors because most of the innovative ingredients considered by such reviewed studies were characterised by multiple different nutritional and functional properties. Accordingly, researchers have developed pasta with various bioactive ingredients, such as:-*Moringa oleifera* pod powder [168];-Purslane [172];-Soy okara [167];-Stinging nettle [169];-Fermented quinoa flour [170];-Potato and pigeonpea flour [178].-Carrob flour [176]-Mushroom powder [171,180];-Brewers spent grain [173,174];-Carrot waste encapsulates [177];-Olive pomace [179];-Fish raw material [165];-Micro and macroalgae [166,175,181,182,183,184].

Research on the above-listed ingredients was carried out by the related authors’ teams with the aim of finding the trade-offs between nutritional improvement and satisfactory sensorial properties of pasta. Those end up confirming durum wheat semolina pasta as an optimal carrier for non-conventional ingredients. This review’s authors pointed out that the addition of ingredients to durum wheat semolina experimented in the studies, although in different proportions (mostly, up to 20% level of replacement), always caused changes in dough, which had various effects on the sensory properties of the product. However, the ingredients were an important source of nutrients, including dietary fibre, minerals, vitamins, phenols, and other bioactive compounds exerting a beneficial impact on human health, for example, phenols.

As a result, the enrichment of pasta products was proven in all of the reviewed studies to be a good strategy for the development of higher-quality pasta, enabling the achievement of both food safety principles, high nutritious values, functionality, and also satisfactory consumer acceptance.

Moreover, some ingredients, such as those obtained from agro-industrial by-products, or those not already consumed to any significant degree, could match the growing request of both the food industry and consumers for more sustainable processes by also providing environmental and economic benefits. For instance, valorising agro-industrial by-products as ingredients for pasta enrichment runs in the direction of circular economy as it allows for minimising waste and generates economic advantages for the agri-food chains involved. Similarly, novel foods such as insects and algae, when added without exceeding the recognised levels for safe ingestion and accordingly to the regulation restrictions (specific for species and food categories) [225], could represent a viable approach to provide pasta with non-meat based proteins of high biological value, along with balanced fatty acid profiles, vitamins, antioxidants, and minerals.

## 4. Conclusions

The authors of the present systematic literature review comprehensively discussed the factors affecting durum wheat quality within the pasta sector, including genotype, cropping, technological, and pasta formulation ones. Effects on nutritional, technological, and health properties of durum wheat grain, semolina, and pasta were explained, along with the improvement options.

Durum-wheat grain composition, which includes macronutrients (proteins, polysaccharides, and lipids), micronutrients (minerals, vitamins), and health components, serves as the starting point for any quality assessment (phytochemicals). The effects of the genotype and growing conditions for the determination of the content and composition of such compounds are of comparable significance, with some traits being more influenced by genotype and others from the environment. Among the others, the bioactive compounds are synthesised by plants as part of their defense mechanism against abiotic and biotic stresses, and their accumulation often increases in response to environmental constraints and the related agronomic management adopted.

Moreover, quality needs to be maintained along the processing steps for the production of semolina and pasta: in that sense, previous research has been quite clear and straightforward in defining which operations should be carefully controlled to obtain high-quality final products (e.g., drying), and which are the novel techniques (e.g., debranning, micronisation) that can contribute to the improvement of quality attributes.

With regard to new pasta formulations, instead, in the last few years, there has been an exponential growth of studies on the addition of alternative ingredients to semolina to produce fortified pasta with enhanced nutritional and functional properties. However, further research on nutrient bioavailability and stability and in vivo antioxidant capacity of these new pasta formulations are needed to successfully design, produce, and include these innovative products in future food habits development.

Accordingly, the review allowed this team of authors to identify a number of promising strategies for improving durum wheat quality from farm to fork, including:Exploring landraces’ germplasm as a source of novel alleles to improve productivity, environmental adaptability, tolerance or resistance to biotic stresses, and nutritional and health value;Making durum wheat nutritional and health-promoting properties the new goals of durum wheat breeding programs;Promoting participatory and decentralised breeding programs to select the desired genotypes also in relation to the choice of the most suitable growing area;Supporting low input practices, including organic farming, such as conservation tillage, crop rotation, and intercropping, to reduce environmental impact, achieve ecosystem services and improve grain quality attributes;Optimising N fertilisation strategies in terms of rate and time to maintain grain technological quality whilst reducing the release of N into the environment;Adopting innovative techniques, such as debranning, air classification, and micronisation, to increase the nutritional and health value of products in the pasta sector;Experimenting with new pasta formulations containing plant-based ingredients in order to contribute to the diversification of healthier and more nutritious diets;Evaluating environmental, nutritional, and health benefits and/or risks associated with new formulations of pasta through holistic and standardised methodologies (e.g., nutritional Life Cycle Assessment), as suggested and carried out in the previously published specialised literature [228,229,230,231].

Thus, the authors concluded that now more than ever, multidisciplinary approaches involving agronomists, breeders, ecologists, food technologists, and economists will be critical for planning more sustainable and high-quality durum wheat production systems [232,233].

## Figures and Tables

**Figure 1 plants-12-00530-f001:**
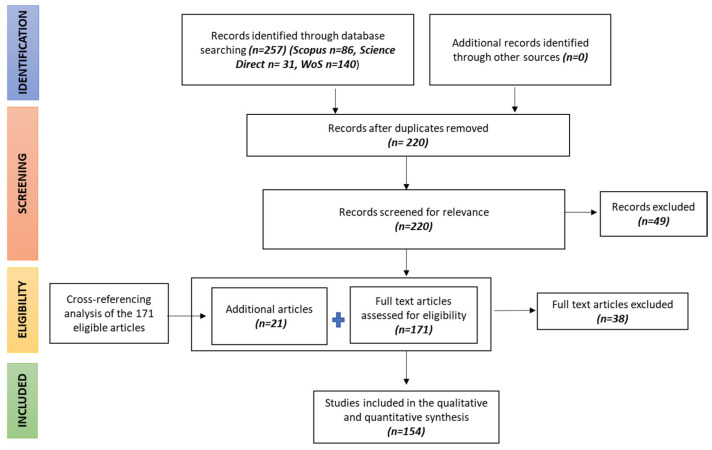
Flowchart of studies included in the present review (*n*- the number of studies), showing the number of articles identified, included, and excluded through the different phases of the systematic review.

**Figure 2 plants-12-00530-f002:**
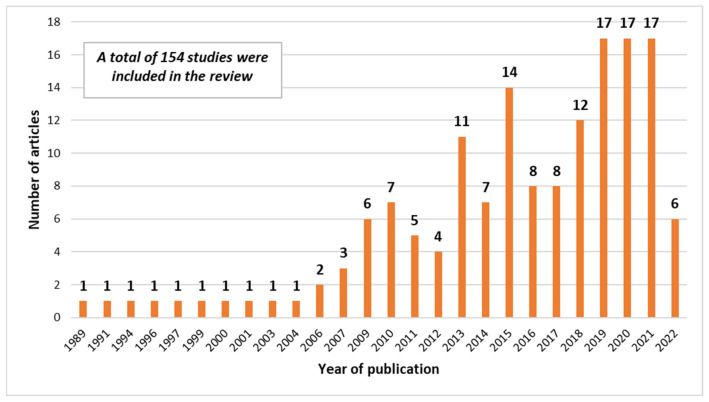
Published papers per year, in the period 1989–2022, according to the review’s criteria.

**Figure 3 plants-12-00530-f003:**
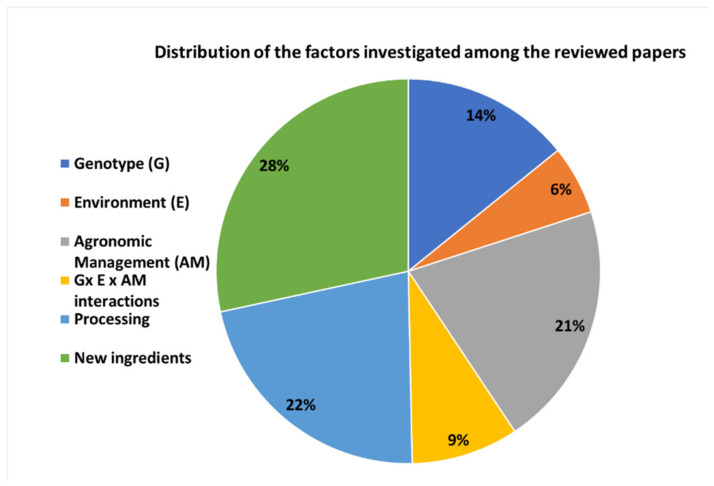
Distribution of the durum-wheat quality factors, as explored by the authors of the papers included in the review sample.

**Table 1 plants-12-00530-t001:** Classification of the reviewed studies according to the type of variable investigated.

Factors’ Class	Factor	Effect on Quality	Study Period ^1^	References
Cropping factors	Genotype (G)	Genotype effects on the yellow and red pigments’ concentration;	2006–2022	[40,41,42,43]
		Genotype effects on the protein and gluten quality and quantity;		[44,45,46,47,48]
		Genotype effects on the macro- and micro-nutrient concentration;		[15,21,49,50,51,52,53,54]
		Genotype effects on the technological quality;		[49,50,51,55,56]
		Genotype effects on the health-promoting compounds’ contents;		[18,20,44,57,58,59,60]
	Environment (E)	Location and climate effects on the technological quality;	1997–2022	[61,62,63]
		Heat and drought stress on the nutritional, technological, and health properties;		[64,65,66,67,68,69,70,71]
		Water stresses on the protein and gluten quantity and quality;		[72,73]
		CO_2_ stresses on the nutritional and technological properties;		[66,74]
	Agronomic Management (AM)	Effects of cropping systems on the nutritional, technological, and health-promoting properties;	2004–2022	[3,75,76,77,78]
		Tillage practices effects on technological quality;		[79,80,81,82,83]
		Fertilisation effects on the nutritional, technological, and health-promoting properties;		[84,85,86,87,88,89,90,91,92,93,94]
		Bio-stimulants’ effects on the nutritional and technological properties;		[95,96,97]
		Herbicides’ effects on the nutritional and technological properties:		[98,99,100]
		Effects of companion and intercropping systems on the protein content and quality;		[101,102]
	G × E × AM	Combined effects of environmental variables, genotype, and/or agronomic management practices on grain and pasta quality attributes;	2011–2020	[19,103,104,105,106,107,108,109,110,111,112,113]
Technological factors	Processing	Effects of fumigation and ozone treatments on the technological properties of grain and semolina;	1989–2021	[114,115]
		Effects of storage conditions and duration on nutritional and technological properties of grain and semolina;		[116,117,118]
		Effects of debranning on the technological and health properties of grain and pasta;		[119,120,121]
		Effects of micronisation and air-classification on the technological and health properties of grain and pasta;		[122,123,124]
		Effects of milling and pasta-making on the nutritional, technological, and health properties;		[117,125,126,127,128,129,130,131,132,133,134,135,136,137,138,139,140,141]
		Effects of germination in reducing gluten peptides associated with CD and on technological features;		[142]
		Effects of drying conditions on the protein denaturation, aggregation and digestibility of pasta;		[143,144,145,146]
Pasta formulation	New ingredients	High-fibre ingredients (bran fractions, psyllium, oat and sorghum flours, beta-glucans, inulin, resistant starch);	2011–2022	[121,147,148,149,150,151,152,153,154,155,156]
		High-protein ingredients (protein concentrates, legume flours, insects);		[157,158,159,160,161,162]
		Probiotics (*Bacillus coagulans* GBI-30);		[163,164]
		Other highly nutritious and functional ingredients;		[165,166,167,168,169,170,171,172,173,174,175,176,177,178,179,180,181,182,183,184]

^1^ Elaborated by this paper’s team of authors, based on the publication years of the reviewed studies.

## Data Availability

No new data were created or analyzed in this study. Data sharing is not applicable to this article.

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
