# Peer review of "Factors Affecting the Nutritional, Health, and Technological Quality of Durum Wheat for Pasta-Making: A Systematic Literature Review"

_plants, 2023, doi:10.3390/plants12030530_

Round 1

Reviewer 1 Report

I carefully revised the paper. In general, I found it relating information about an interesting and actual subject, well written and informative. However, before considering for publication, some shortcomings should be addressed.

Most of the review is not directly related to pasta, but to durum wheat as a raw material. I think that the title should be changed considering this aspect (“…durum wheat for pasta…”) and the revision of literature should be completed, if the item “pasta” is kept in the title. The part regarding pasta processing is short and incomplete (consider https://doi.org/10.1016/j.jcs.2010.06.002; https://doi.org/10.1016/j.foodchem.2010.01.034; https://doi.org/10.1016/j.foodchem.2011.09.034). Enzyme activity is an important aspect (consider https://doi.org/10.1016/j.foodchem.2011.09.132  and https://doi.org/10.1016/j.jcs.2013.04.004) on processing and quality (color, antioxidants, heat damage: https://doi.org/10.1016/j.foodcont.2021.108036; 10.1016/j.foodchem.2013.10.071; https://doi.org/10.1016/j.lwt.2020.109932; Boggini et al. 1999, Effect of durum wheat genotype and environment on the heat-damage of dried pasta, Journal of Genetics and Breeding, 53, 337 – 347), but is not considered at all. As stated by the Authors color is an important aspect of pasta; thus carotenoid content (https://doi.org/10.3390/foods10040757) and its stability during pasta making and cooking are fundamental issues, not considered in this review (https://doi.org/10.1080/09637486.2022.2029831 and others).

Lines 258-262. Also granulometry determine the color (10.1016/j.foodres.2014.06.050), it must be mentioned in some part of the review.

Line 850. What about mixing? It is a critical step of the process because of enzyme activities (see above).

Lines 902-934. The text should be moved to line 849. Also in this issue several studies were excluded (for example, https://doi.org/10.1016/j.foodres.2016.05.012).

The English language is good.

Author Response

Please, see the attachment with the reply.

Reviewer 2 Report

1. Congrats on the Flowchart of studies - Figure 1. Flowchart of studies included in the present review (n- the number of studies), 116 showing the number of articles identified, included, and excluded through the different phases of 117 our systematic review.  

2. Congratulations for this Review, it is complex, it contains a significant number of bibliographic titles 176, I suggest you also make an abstract graphic, where an overview of the entire Review will appear.

Author Response

(The authors gave the same response as above.)

Reviewer 3 Report

plants-2030599

The manuscript deals with an interesting review of the literature about the factors affecting the nutritional, health and technological quality of durum wheat pasta sector products. It includes cropping factors, technological factors and pasta formulations.

However, and if it is a systematic literature review according to the PRISMA flow chart, there are some works non include here, namely about the incorporation of microalgae and macroalgae in food products. Which were the criteria?

Major concerns:

In the Introduction, Authors only refer to the importance of gluten, but something should be said about starch and its nutritional and technological importance.

In my opinion, concerning agronomic management effects, is very important to mention the glyphosate controversy and include papers about its effects on human health.

When talking about algae and insects, Authors must refer the legal limitations.

Minor changes: Please see my comments in the attached pdf file.

Author Response

(The authors gave the same response as above.)

Reviewer 4 Report

Very interesting approach to the topic, the work can be a good source material. Knowledge has been systematized.
I have No comments. I recoment to publish.

Author Response

(The authors gave the same response as above.)

Round 2

Reviewer 1 Report

The Authors improved the paper as suggested.

Reviewer 3 Report

In my opinion, the present version of the paper could be accepted for publication. The authors considered all the sugestions made by the reviewers.